# Genomic and Metabolomic Analyses of a Piezosensitive Mutant of *Saccharomyces cerevisiae* and Application for Generation of Piezosensitive Niigata-*Sake* Yeast Strains

**DOI:** 10.3390/foods10102247

**Published:** 2021-09-23

**Authors:** Toru Shigematsu, Yuta Kaneko, Minami Ikezaki, Chihiro Kataoka, Kazuki Nomura, Ayana Nakano, Jotaro Aii, Toshio Aoki, Takashi Kuribayashi, Mitsuoki Kaneoke, Saori Hori, Akinori Iguchi

**Affiliations:** 1Faculty of Applied Life Sciences, Niigata University of Pharmacy and Applied Life Sciences, 265-1 Higashijima, Akiha-ku, Niigata 956-8603, Japan; yutakaneko@nupals.ac.jp (Y.K.); chihirokataoka@nupals.ac.jp (C.K.); k_nomura@neptune.kanazawa-it.ac.jp (K.N.); ayana_nakano@nupals.ac.jp (A.N.); jotaroaii@nupals.ac.jp (J.A.); ujiie@nupals.ac.jp (S.H.); a_iguchi@nupals.ac.jp (A.I.); 2Graduate School of Applied Life Sciences, Niigata University of Pharmacy and Applied Life Sciences, 265-1 Higashijima, Akiha-ku, Niigata 956-8603, Japan; minamiikezaki@nupals.ac.jp; 3Department of Applied Bioscience, College of Bioscience and Chemistry, Kanazawa Institute of Technology, 7-1 Ohgigaoka, Nonoichi 921-8501, Japan; 4Niigata Prefectural Sake Research Institute, 2-5932-133 Suidocho, Chuo-ku, Niigata 951-8121, Japan; aoki.toshio@pref.niigata.lg.jp (T.A.); takashi-kuribayashi@nafu.ac.jp (T.K.); kaneoke.mitsuoki@pref.niigata.lg.jp (M.K.); 5Food Industry Department, Niigata Agro-Food University, 2416 Hiranedai, Tainai 959-2702, Japan

**Keywords:** *sake* (Japanese rice wine), high hydrostatic pressure (HHP) processing, pasteurization, *Saccharomyces cerevisiae*, *sake* yeast, pressure-sensitive (piezosensitive) mutant, genome analysis, metabolome analysis

## Abstract

A sparkling-type draft cloudy *sake* (Japanese rice wine), AWANAMA, was recently developed using high hydrostatic pressure (HHP) treatment as a non-thermal pasteurization method. This prototype *sake* has a high potential market value, since it retains the fresh taste and flavor similar to draft *sake* while avoiding over-fermentation. From an economic point of view, a lower pressure level for HHP pasteurization is still required. In this study, we carried out a genome analysis of a pressure-sensitive (piezosensitive) mutant strain, a924E1, which was generated by UV mutagenesis from a laboratory haploid *Saccharomyces cerevisiae* strain, KA31a. This mutant strain had a deletion of the *COX1* gene region in the mitochondrial DNA and had deficient aerobic respiration and mitochondrial functions. A metabolomic analysis revealed restricted flux in the TCA cycle of the strain. The results enabled us to use aerobic respiration deficiency as an indicator for screening a piezosensitive mutant. Thus, we generated piezosensitive mutants from a Niigata-*sake* yeast strain, S9arg, which produces high levels of ethyl caproate but does not produce urea and is consequently suitable for brewing a high-quality *sake*. The resultant piezosensitive mutants showed brewing characteristics similar to the S9arg strain. This study provides a screening method for generating a piezosensitive yeast mutant as well as insight on a new way of applying HHP pasteurization.

## 1. Introduction

*Sake* (Japanese rice wine) is produced via the multiple parallel fermentation process, which consists of the simultaneous saccharification of rice starch by *Aspergillus oryzae* (*koji* mold) enzymes and ethanol fermentation by *Saccharomyces cerevisiae* (*sake* yeast), which is subsequently subjected to thermal pasteurization before the shipment. Unpasteurized draft *sake* has a potentially high market value due to its fresh flavor and fruity taste compared to thermal-pasteurized *sake*, although draft *sake* has a short shelf life resulting from the deterioration of flavors and taste due to over-fermentation by *sake* yeast and the activity of the remaining enzymes produced by *koji*-mold. We conducted studies examining the inactivation of yeasts under high hydrostatic pressure (HHP) [1,2,3] in order to develop an HHP application to suppress the over-fermentation of fermented foods, known as the pressure-regulated fermentation (i.e., PReF) technology [4].

Recently, we developed a sparkling-type draft cloudy *sake*, AWANAMA, with HHP treatment as a non-thermal pasteurization method [5,6]. This prototype *sake* retains a fresh taste and flavor similar to draft *sake*, while avoiding over-fermentation by HHP treatment at 400 MPa and at ambient temperatures for 10 min. However, from an economic point of view, further improvements to the brewing process for commercialization still require new techniques which will enable sufficient HHP pasteurization at lower pressure levels. For this purpose, the generation of pressure-sensitive (piezosensitive) *sake* yeast strains is needed.

We previously obtained a piezosensitive mutant strain, a924E1, via ultraviolet (UV) mutagenesis of a budding yeast *S. cerevisiae* strain KA31a [1]. Strain a924E1 showed a larger loss of viability than the parent strain when subjected to HHP treatment at 175 to 250 MPa at temperatures from 4 to 40 °C [7], as well as when subjected to thermal treatment at 50 to 58 °C under atmospheric pressure [2]. These results indicated that this mutant strain was sensitive to both HHP and thermal stresses. Transcriptome analysis, as well as phenotypic analysis, of this mutant strain suggested that the strain had deficient aerobic respiration and mitochondrial functions [8]. Moreover, PCR amplification targeting genes in the mitochondrial DNA suggested that a region of the *COX1* gene, encoding the subunit I of cytochrome *c* oxidase, in the mitochondrial DNA was deleted in strain a924E1. However, the relationship between the dysfunction of the *COX1* gene and the increased piezosensitivity is still not fully understood.

In this study, we analyzed the genome sequence of the piezosensitive strain a924E1 and compared with that of the parent strain KA31a. To analyze the mechanism of increased piezosensitivity, metabolic profiles of the energy metabolism, glycolysis, and the tricarboxylic acid (TCA) cycle of strain a924E1 were analyzed and compared to those of the parent strain KA31a. Furthermore, we generated piezosensitive mutants from a Niigata-*sake* yeast strain S9arg for an application study. Strain S9arg, used as the parent strain, is a high ethyl-caproate producing and non-urea producing strain, which was derived from the original strain G9 [9], which produces high levels of ethyl caproate but does not produce urea, and is thus suitable for brewing a high-quality *sake*.

## 2. Materials and Methods

### 2.1. Yeast Strains and Growth Conditions

Yeast strains used were the wild-type haploid *S. cerevisiae* strain KA31a (*MATa*, *his3*, *leu2*, *trp1*, and *ura3*) and its piezosensitive mutant strain a924E1 (*MATa*, *his3*, *leu2*, *trp1*, and *ura3*) [1]. The Niigata-*sake* yeast *S. cerevisiae* strain S9arg used in this study was provided by the Niigata Prefectural Sake Research Institute (Niigata city, Niigata, Japan). The piezosensitive strains derived from strain S9arg, strain UV1, and NM1, were produced during this study. The strains were usually grown aerobically on a YPD medium (1% yeast extract, 2% peptone; Becton Dickinson and Co., Franklin Lakes, NJ, USA, and 2% glucose; FUJIFILM Wako Pure Chemical, Osaka, Japan) at 30 °C for 48 h.

### 2.2. Screening of Aerobic Respiration-Deficient Mutants

A cell suspension of strain S9arg was subjected to UV irradiation treatment in accordance with to our previous report [1]. The cell suspension after UV irradiation treatment was serially diluted with 0.85% NaCl solution and plated on a YPD medium containing 1.5% agar. After cultivation at 30 °C for 48 h, the colonies on the YPD agar medium were subjected to a TTC staining test to screen the aerobic respiration-deficient mutant in accordance with the description by Nagai [10]. A TTC-agar medium (0.5% glucose, 0.05% 2,3,5-triphenyl-2*H*-tetrazolium chloride (TTC; FUJIFILM Wako Pure Chemical), and 1% agar) was overlayed onto the YPD agar medium. After approximately 30 min, the color of colonies with aerobic respiration function change to red (TTC-positive). The colonies displaying a white color (TTC-negative) were regarded as aerobic respiration-deficient mutants. A cell suspension of strain S9arg without UV irradiation treatment was also subjected to a TTC staining test.

### 2.3. High Hydrostatic Pressure (HHP) Treatment

HHP treatment was carried out in accordance with our previous report [1]. The cultivated yeast cells were collected and diluted 10-fold with sterilized 0.85% NaCl solution. The cell suspension was vacuum-packed in polyethylene bags and soaked in distilled water (pressure-medium) in the stainless-steel vessel of an HHP apparatus (WIP, Kobe Steel, Kobe, Japan). HHP treatment was performed at 200 MPa at an ambient temperature (around 20 °C).

The HHP inactivation of yeast cells was evaluated by viable cell counts using the colony-count method. The cell suspensions after HHP treatment were serially diluted with sterilized 0.85% NaCl solution. For calculation of the viable cell count, these cell suspensions were plated on a Compact Dry Nissui YM dry sheet medium culture plate (Nissui Pharmaceutical, Yuuki, Japan) and cultivated at 30 °C for 3 days. The viable cell numbers were calculated based on the number of colonies that appeared.

### 2.4. Genome Analysis

The total DNA was extracted after 48 h of cultivation of the yeast strain. The sequencing of genome DNA from strain KA31a was performed using GridION X5 (Oxford Nanopore Technologies, Oxford, UK) and the HiSeq 2500 (Illumina) instrument at GeneBay, Yokohama, Japan. The hybrid de novo assembly with Nanopore long-read and Illumina short-read was performed by Canu v.1.7.1, smartdenovo, and Unicycler v.0.4.6. The genome polishing and finishing was performed using Pilon v.1.22 and Burrows-Wheeler Aligner (BWA) v.0.7.17. To evaluate completeness of the gene set in the assembled contigs, we applied bench-marking universal single copy (BUSCO) analysis v.3.0.2. For the BUSCO analysis, we set “genome” as the assessment mode and used stain S288C as the database, and obtained 98.2% complete BUSCOs. A dot-plot analysis was performed using MashMap with the default settings.

The sequencing of the genome DNA of strain a924E1 was analyzed using Hiseq 2500. The obtained reads were mapped to assembled contigs of strain KA31a by BWA v.0.6.1. Variant calling was performed using Samtools v.0.1.18, Bcftools v.0.1.18, and Genome Analysis Toolkit (GATK) v4.1.0.0. These variations were further checked manually using the Integrative Genomics Viewer (IGV) v.2.4.3.

The genome sequence data of the reference strains S288C and K7 (Kyoukai No. 7) were obtained via the *Saccharomyces* Genome Database (SGD) by Stanford University, USA (https://www.yeastgenome.org/ (accessed on 31 August 2021)) and the *sake* yeast genome database (SYGD) by the National Research Institute of Brewing, Japan (https://nribf1.nrib.go.jp/SYGD/ (accessed on 31 August 2021)), respectively.

### 2.5. PCR Analysis

PCR amplification targeting the *COX1* gene region was carried out in order to analyze the deleted region in the mutant strains. The total DNA extracted from yeast cells was used as the templates. The primers used are shown in Table 1. The amplified products were analyzed by agarose gel electrophoresis.

For the analysis of other genes on the mitochondrial DNA of strain S9arg and its piezosensitive mutants, the PCR primer sets targeting the 21S rRNA, *COX3*, *COB*, and *ORI7* genes were used. The primer set targeting the 21S rRNA gene was mt59F, 5′-CGCACTTTGCAGAAACGATA-3′ and mt59R, 5′-ATAATGACGCCCCATCAAAA-3′. The primer set targeting *COX3* was mt80F, 5′-TCTTTGCTGGTTTATTCTGAGC-3′ and mt80R, 5′-TACCTGCGATTAAGGCATGA-3′. The primer set targeting *COB* was mt36F, 5′-GGGTTCTTTGATGCTGATGG-3′ and mt36R, 5′-TATGGGAGTTCCCACAAAGC-3′. The primer set targeting *ORI7* was mt29F, 5′-GACCTCACTCCTTCCCCACT-3′ and mt29R, 5′-CCCTCCCCCTATTACGTCTC-3′.

For the analysis of the polyploidy of strains S9arg and its piezosensitive mutants, an analysis of the *MAT* locus was carried out via colony PCR using a KODFX kit (KFX-101; TOYOBO, Osaka, Japan) with the *MAT*-specific primer 5’-AGTCACATCAAGATCGTTTATGG-3’, the *MATa*-specific primer 5’-ACTCCACTTCAAGTAAGAGTTTG-3’, and the *MATα*-specific primer 5’-GCACGGAATATGGGACTACTTCG-3’ in accordance with a report by Huxley et al. [11].

### 2.6. Extraction and Analyses of Metabolites

At time points of 6, 12, 24, and 48 h of cultivation, intracellular metabolites were extracted from the cells as described by Hasunuma et al. [12]. Briefly, a 5 mL portion of the culture broth was mixed with 7 mL of pre-chilled (−60 °C) 60% (*v/v*) methanol in water. The mixture was centrifuged at 5000× *g* for 5 min at −9 °C and the supernatant was discarded. The pellet was resuspended in 80 µL of 1.25 mM 1,4-piperazinediethanesulfonic acid (PIPES), which was used as the internal standard. To extract the metabolites, 75% (*v/v*) ethanol, preheated to 95 °C in an aluminum block bath (model DTU-1C, TAITEC, Koshigaya, Japan), was rapidly added to the resuspended cell pellet. The mixture was immediately vortexed, and the sample was placed in the aluminum block bath at 95 °C for 3 min. After cooling on ice, the extracts were evaporated at 40 °C using a centrifugal evaporator (model CVE-3100, Tokyo Rikakikai, Tokyo, Japan). The dried samples were resuspended in 100 µL of milli-Q water. After centrifugation at 20,400× *g* at 4 °C for 5 min (model MX-301, Tomy Seiko, Tokyo, Japan), the supernatants were used for further analysis.

An analysis of the metabolites was performed via capillary electrophoresis-electrospray ionization-mass spectrometry (CE-ESI-MS) using a modified version of the method employed by Harada et al. [13]; the process is summarized below. CE-ESI-MS analyses were performed using a P/ACE MDQ capillary electrophoresis system (SCIEX, Framingham, MA, USA) and a 3200 QTRAP LC/MS/MS system (SCIEX). The CE separations were performed using an 80 cm length of capillary tubing (PN 338472, SCIEX) with an inner diameter (I.D.) of 50 µm and an outer diameter (O.D.) of 375 µm. The electrolyte for the CE separation was 50 mM ammonium acetate solution. Before the injection for each analysis, the capillary was pretreated with a running electrolyte for 2 min at 5 psi (approximately 34.5 kPa). The sample then was injected at a pressure of 5 psi for 10 s, followed by the injection of a running electrolyte at 5 psi for 5 s. The applied voltage for separation was + 30 kV for 8 min. The voltage application was stopped after 8 min, and the electrolyte then was delivered through the capillary at 4.5 psi (approximately 31.0 kPa) for 13 min. The capillary was maintained at 20 °C throughout the process. The sheath liquid (5 mM ammonium formate in 50% (*v/v*) acetonitrile/water) was delivered to the electrospray probe at a rate of 5 µL/min. ESI-MS/MS was conducted in the negative ion mode. An ion spray voltage was applied at −4.5 kV starting 1 min after the start of the voltage application to the CE. The metabolites in samples separated by CE were detected by MS/MS in the multiple reaction monitoring (MRM) mode. The measurement parameters of ESI-MS/MS for each analyzed metabolite were optimized using the Analyst software (SCIEX); the resultant Q1 (m/z of deprotonated precursor ion) and Q3 (m/z of the fragmented ion) are listed in Table 2.

### 2.7. Laboratory-Scale Sake Brewing Test

A laboratory-scale *sake* brewing test was carried out based on our previous report [14]. Yeast strains were pre-cultivated in 3 mL of YPD medium at 30 °C for 24 h with shaking. After pre-cultivation, 160 μL aliquot of the culture broth was inoculated into 200 mL of YPD medium containing 10% glucose and cultivated at 15 °C for 48 h without shaking. The culture broth was then transferred into a 250 mL volume NALGENE bottle (Thermo Fisher Scientific, Waltham, MA, USA) and the yeast cells were collected by centrifugation at 3000 rpm for 5 min at 4 °C (Model 6200, (Kubota, Tokyo, Japan). The viable cells after centrifugation were counted using a staining method with 0.4% trypan blue solution (FUJIFILM Wako Pure Chemical). A 4.8 mL of cell suspension containing 8.4 ± 0.1 log cells of the strains was used as the fermentation starters (*shubo*). The water *koji* was prepared by adding 50 mL of sterilized water to dry malted rice and pre-cultivated at 15 °C for 24 h without shaking. The *shubo* and a 10% lactic acid solution were followed by steamed rice, dry malted rice, and sterilized water divided into three portions, and added to prepare the fermentation mash (*moromi*). This traditional mashing method in three stages is called a *sandan-jikomi*, and the stages of adding materials are called a *hatsu-zoe* (first addition), a *naka-zoe* (second addition), and a *tome-zoe* (third addition), respectively. The brewing temperature of the *moromi* was controlled at 15 °C in the *hatsu-zoe*, at 9 °C in the *naka-zoe*, and at 7 °C in the *tome-zoe* for 1 day, respectively. The day of the *tome-zoe* (third addition of materials) was defined as Day 0 of brewing, and then increased by 1 °C per day from 7 to 15 °C. After Day 8, the brewing temperature was controlled at 15 °C until the end of the brewing period. The brewing behavior was evaluated by a decrement in volume of the *moromi* weight, due to CO_2_ output during ethanol fermentation.

### 2.8. Characterization of Flavor Components in the Moromi

For the characterization of the brewing properties, the supernatant of the *moromi* was harvested via centrifugation at 12,000 rpm for 10 min at 4 °C (Model 6200, Kubota, Tokyo, Japan) in accordance with a previous report [14]. The concentrations of flavor compounds were analyzed by GC/MS system (GCMS-QP2010 SE; Shimadzu, Kyoto city, Kyoto, Japan) with a headspace sampler (HS-20; Shimadzu, Kyoto, Japan). The GC/MS analyses were performed using a gas chromatograph (GC-2010 Plus) directly connected to a mass spectrometer. This GC/MS system was equipped with an InertCap Pure Wax column (0.25 mm i.d. × 30 m length × 0.25 μm film thickness; GL Sciences, Tokyo, Japan) and helium was used as a carrier gas. The column temperature was held at 45 °C for 5 min, increased from 45 to 240 °C at 3 °C per min, and then held at 240 °C for 5 min. The load volume using the headspace sampler was 1 μL. The compounds separated by the GC were ionized through electron impact ionization (EI) and the ion source temperature was 200 °C. The interface temperature was 230 °C with scanning from 29.00 to 400.00 m/z. The MS spectra of the separated compounds were compared with one from the Smart Metabolites Database and analyzed using Lab Solutions Insight (ver. 3.1) software (Shimadzu, Kyoto, Japan).

## 3. Results and Discussion

### 3.1. Growth and Pressure Inactivation Characteristics of Strains KA31a and a924E1

Figure 1a shows the typical growth curves of strains KA31a (wildtype) and its pressure sensitive (piezosensitive) mutant strain a924E1, grown at 30 °C in a YPD medium. Both strains showed comparable growth characteristics within 24 h of cultivation. From 24 h to 60 h after cultivation, strain a924E1 showed a lower cell concentration compared with strain KA31a. The pressure inactivation characteristics 48 h after cultivation, when both strains were in a stationary phase, are shown in Figure 1b. Strain a924E1 appeared to demonstrate a larger loss of viability compared with strain KA31a. This result showed the increased piezosensitivity of strain a924E1, which agreed with our previous reports [1,2,3].

### 3.2. Genome Analysis of Strains KA31a and a924E1

#### 3.2.1. Genome Analysis of Strain KA31a

The whole genome of strain KA31a was analyzed. The total DNA was extracted from strain KA31a and subjected to next-generation sequencing using an Illumina sequencer and a Nanopore sequencer. Based on the sequence data obtained, the chromosomal DNA sequence was assembled de novo. Based on the Illumina sequencing, 14,484,172 reads with an average length of 146 bp, which covered 2108 Mbp in total bases, were obtained. Based on the Nanopore sequencing, 1,320,882 reads with an average length of 6351 bp, which covered 2108 Mbp in total bases, were obtained. Through hybrid assembly, 18 contigs with an average size of 669,105 bp and a 1,544,297 bp maximum were obtained. The total contig size was calculated to be 12,043,888 bp. The *S. cerevisiae* strain S288C genome, which was used as a reference genome sequence, consists of 16 chromosomes. Fourteen contigs in the 18 contigs obtained from strain KA31 were annotated to Chromosome I to XI and XIV to XVI in strain S288C. In the remaining four contigs from strain KA31a, two contigs and two contigs were annotated to chromosome XII and chromosome XIII in strain S288C, respectively. The chromosome number of strain KA31a was considered to be 16, as in strain S288C.

The mitochondrial and plasmid DNA sequence was also assembled. Through the Illumina sequencing, 1,447,540 reads with an average length of 146 bp, which covered 210 Mbp in total bases, were obtained. By using the Nanopore sequencing, 17,035 reads with an average length of 6601 bp, which covered 114 Mbp in total bases, were obtained. Through hybrid assembly, a contig of 88,479 bp in size was detected, which was annotated to the mitochondrial DNA. A contig of 6193 bp in size was also detected, which was annotated to the 2 micron plasmid *S. cerevisiae*.

The genome sequence of strain KA31a was compared with that of strain S288C (Figure 2). The overall sequence was well-conserved between the two strains. However, on chromosome XIV in strain S288C, an inversion region of approximately 30 kbp (corresponding nucleotide number 602 to 570 in strain S288C) was detected in strain KA31a. This inversion was also detected between chromosome XIV of strain S288C and *S. cerevisiae* strain K7 (Kyoukai No. 7), which is a popular *sake* yeast. The genome structure of strain KA31a was indicated to be more closely related to strain K7 than to strain S288C.

#### 3.2.2. Genome Analysis of Strain a924E1

A comparison analysis of the genome sequence of strain a924E1 with strain KA31a was carried out. The total DNA of strain a924E1 was extracted and subjected to Illumina sequencing. The sequences of obtained short reads were mapped against the genome sequence of strain KA31a. A large deletion of 10,130 bp in mitochondrial DNA was detected in strain a924E1 (Figure 3). The deleted region included the region between the second exon and the eighth exon of the *COX1* gene. Additionally, we carried out PCR amplification of this region using five primer sets covering the whole deleted region. None of the five primer sets could provide amplification. Our previous results, in which this gene was not detected by a PCR experiment, [8] is thus attributable to the deletion of this region. No other genes or open reading frames were detected within the deleted region.

Our comparison analysis revealed a total of 2147 single nucleotide polymorphisms (SNPs) between strain a924E1 and strain KA31a. Using the GATK tool, the 2147 SNPs were narrowed down to four SNPs. The regions containing these four SNPs were PCR amplified and sequenced. These SNPs were confirmed to exist between the two strains. Two SNPs were shown to be silent mutations. The remaining two SNPs were missense mutations in the *RRG1* and *CDC37* genes. The mutation in *RRG1* is reported to cause deficiencies in maintaining the mitochondria [15]. The *RRG* gene family regions (*RRG1*, *RRG8* and *RRG9*) of the piezosensitive mutant strains UV1 and NM1 derived in this study from the Niigata-*sake* yeast strain S9arg were PCR amplified and sequenced. None of the strains tested showed mutations in the *RRG* genes. The SNPs detected in strain a924E1 thus did not appear to be related to the increased piezosensitivity in strain a924E1. We thus concluded that the increased piezosensitivity in strain a924E1 was caused by the deletion of the *COX1* gene.

### 3.3. Metabolites Analysis of Strains KA31a and a924E1

#### 3.3.1. Metabolites of Glycolysis

Strains KA31a (wild-type) and a924E1 (piezosensitive mutant) were cultivated on a YPD medium at 30 °C for 48 h. At time points of 6, 12, 24, and 48 h during cultivation, the culture broths were sampled and analyses of the metabolites were performed. The cultivation times of 6, 12, 24, and 48 h corresponded to the middle of the logarithmic growth phase, the end of logarithmic growth phase, the early stationary phase, and the late stationary phase, respectively (Figure 1a).

The glycolysis metabolites of strains KA31a and a924E1 during cultivation were analyzed (Figure 4). For both strains, the concentration of intracellular glucose (and possibly fructose, because glucose and fructose could not be distinguished) after 6 h of cultivation was approximately 2000 nmol/g (dry cell weight), and increased to approximately 3000 nmol/g at 12 h (Figure 4a).

The concentrations of intracellular glucose decreased thereafter, such that only trace levels of glucose were detected after 24 h of cultivation. During the logarithmic growth phase, cells were uptaking and accumulating glucose. The end of the logarithmic growth phase corresponds to the depletion of glucose from the medium. The intracellular glucose was consumed during the interval of 12 to 24 h. No apparent differences in glucose concentrations were observed between the two strains during the 48 h period of cultivation, indicating that there were no differences between the strains in glucose uptake from the medium or in glucose conversion to glucose-6-phosphate via glycolysis.

The concentrations of other metabolites in the glycolysis, glucose-6-phosphate (glucose-1-phosphate may also be present due to the inability to distinguish between glucose-6-phosphate and glucose-1-phosphate), fructose-1,6-bisphosphate, 3-phosphoglycerate and phosphoenolpyruvate, also showed similar patterns in the two strains (Figure 4b–e). As with glucose, the concentrations of these other metabolites initially increased with the cultivation time, reaching a maximum at 12 h before decreasing to trace levels after 24 h of cultivation.

These results suggest that glycolysis proceeds similarly in the two strains. It should be noted that the concentrations of fructose-1,6-bisphosphate and phosphoenolpyruvate were significantly lower in strain a924E1 than in strain KA31a (Figure 4c,e). Although glycolysis proceeded similarly in both strains, some restrictions appeared to be present in the production of fructose-1,6-bisphosphate and phosphoenolpyruvate in strain a924E1, presumably due to the deletion of the *COX1* gene. The concentrations of the electron carriers nicotinamide adenine dinucleotide (NAD^+^) and reduced nicotinamide adenine dinucleotide (NADH) were also significantly lower in strain a924E1 than in strain KA31a (Figure 4f,g). The reduction of NAD^+^ to NADH occurred during the oxidation of glyceraldehyde-3-phosphate to 1,3-bisphophoglycerate during glycolysis. The oxidation of NADH to NAD^+^ occurred during the reduction of acetaldehyde to ethanol in the final step of ethanol fermentation.

These results suggest that some restrictions exist during these steps in strain a924E1. However, our previous study indicated that strain a924E1 has an ethanol fermentation ability equivalent to that of strain KA31a [1]. Thus, the restrictions observed in the present study in the glycolytic and ethanol fermentation pathways in strain a924E1 did not appear to cause a restriction effect on the overall flux in these pathways. Notably, the deletion of the *COX1* gene was expected to affect mitochondrial functions, without directly affecting cytosol-located glycolysis and the ethanol fermentation pathway. The deduced flux of the metabolites revealed by the present study is illustrated in Figure 5. The metabolic pathways in *S. cerevisiae* portrayed in Figure 5 are based on the PATHWAY database of the Kyoto Encyclopedia of Genes and Genomes (KEGG) (http://www.genome.jp/kegg/ (accessed on 31 August 2021)).

#### 3.3.2. Metabolites in the TCA Cycle

Next, the metabolites in the tricarboxylic acid (TCA) cycle of strains KA31a and a924E1 were analyzed. For strain KA31a, the concentrations of all metabolites analyzed (including citrate, malate, 2-oxoglutarate, succinate, acetyl-CoA, and fumarate) initially increased with cultivation time, achieving a maximum concentration at 12 h before a subsequent gradual decrease during the 48 h time period (Figure 6). During the first 12 h of cultivation time, a portion of the pyruvate produced from glucose via glycolysis was expected to be consumed by the TCA cycle. After glucose became limiting at 12 h, the TCA metabolites were presumably progressively consumed during the 48-h time point.

For strain a924E1, the concentrations of citrate and malate initially increased with cultivation time, producing the maximum concentrations at 12 h of cultivation time, and then gradually decreased from 12 h to 48 h. This pattern of behavior was similar to those observed with strain KA31a (Figure 6a,b). However, the concentrations of 2-oxoglutarate and succinate in strain a924E1 showed a different pattern of behavior from those observed with strain KA31a (Figure 6c,d). Specifically, concentrations of 2-oxoglutarate and succinate initially increased gradually, producing the maximum concentrations at 24 h of cultivation time, and then decreased by 48 h. Moreover, in strain a924E1, only trace levels of acetyl-CoA and fumarate were detected during all cultivation times (Figure 6e,f).

The trace levels of acetyl-CoA detected in strain a924E1 suggested an almost complete limitation of acetyl-CoA production. Nevertheless, the concentration of citrate in strain a924E1 was comparable to that in KA31a. Thus, the trace level of acetyl-CoA was not considered to reflect a change in citrate production. Notably, the concentration behaviors of 2-oxoglutarate and succinate in strain a924E1 exhibited an approximately 12-h delay compared to those in strain KA31a. This result suggested that the production of 2-oxoglutarate and succinate in strain a924E1 was restricted. Additionally, fumarate production was restricted in strain a924E1, suggesting that the conversion of succinate to fumarate was inhibited. Surprisingly, the concentration behavior of malate in strain a924E1 was comparable to that in strain KA31a. The production of malate in strain a924E1 may occur via pyruvate and oxaloacetate and/or from isocitrate via the glyoxylate cycle.

The deduced flux of the metabolites revealed by this study is illustrated in Figure 5. In strain a924E1, which harbors a deletion of the *COX1* gene, a significant restriction was found to occur in the metabolic conversion of 2-oxoglutarate to fumarate. Notably, the production of fumarate from succinate was significantly restricted. This reaction step is catalyzed by the enzyme succinate dehydrogenase (EC 1.3.5.1), which is located on the inner membrane of the mitochondrion and forms complex II of the respiration chain. This restriction on the TCA cycle in strain a924E1 might cause an alteration of the metabolite flux in order to permit the production of malate from pyruvate via oxaloacetate by pyruvate carboxylase (EC 6.4.1.1) and malate dehydrogenase (EC 1.1.1.37). Another possible mechanism of malate production is an alteration of the metabolic flux to the glyoxylate cycle, whereby isocitrate is converted to succinate and malate during the conversion of acetyl-CoA to HS-CoA, reactions which are catalyzed by isocitrate lyase (EC 4.1.3.1) and malate synthase (EC 2.3.3.9). Although a more detailed analysis of the metabolic flux in a924E1 is still needed, the present study indicated that the mutant strain is restricted in the TCA cycle, presumably as a result of the deletion of the *COX1* gene.

#### 3.3.3. Relationship between the Restricted Metabolism and Piezosensitivity

In general, microbial inactivation by HPP was considered to result from damage inflicted to the structures and functions of the cell and organelle membranes. Transmission electron microscopic observation of *S. cerevisiae* cells revealed that the nuclear membrane is damaged by HHP treatments over 100 MPa [16]. The function of amino acid transporters located in the cellular membrane of *S. cerevisiae* is also impaired by pressure treatment [17,18]. Using an *S. cerevisiae* deletion library, it was shown that ergosterol was required for growth to occur under conditions of HHP at 25 MPa and a temperature of 15 °C [19]. Increases in the concentration of trehalose, a protectant of cellular membranes, was shown to provide increased piezotolerance [20]. Moreover, pressure inactivation of microorganisms is associated with damage to the cytoskeletal protein complex and with impaired function of the cellular systems responsible for degrading misfolded proteins [21].

Nanba et al. compared the growth ability of the piezosensitive *S. cerevisiae* strain a924E1, as well as a piezotolerant strain, after HHP treatment [3]. Those authors observed a growth delay in strains after HHP treatment and suggested that the recovery functions were important for survival following cellular damage by HHP treatment. Thus, a strain with enhanced recovery functions was expected to show increased piezotolerance, and a strain with impaired recovery functions was expected to exhibit increased piezosensitivity. In the present study, the restriction of the TCA cycle flux in the piezosensitive *S. cerevisiae* strain a924E1 was revealed through metabolomic analysis. Since extracellular glucose was exhausted after 12 h cultivation on a YPD medium, energy generation, especially after 24 h of cultivation, would depend on the TCA cycle, possibly via a process linked to the aerobic respiratory chain. In the wild-type strain KA31a, metabolites of the TCA cycle were still detected even at 48 h of cultivation, suggesting flux through the TCA cycle was still operational at this time point. In contrast, the mutant strain a924E1 showed a deficiency in the TCA cycle, which was expected to lead to a lack of the energy generation needed for recovery from cellular damage.

Nomura et al. employed metabolomic analysis to demonstrate that arginine synthesis is significantly lower in strain a924E1 compared to strain KA31a [22]. The authors confirmed that arginine supplementation of the medium enhanced the survival of strain a924E1 after HHP treatment. Arginine biosynthesis originates from 2-oxoglutarate, a metabolite in the TCA cycle. Thus, the restricted TCA cycle of strain a924E1 would relate to other TCA-dependent metabolic processes, such as amino acid synthesis. The present study thus demonstrates the importance of the function of the TCA cycle for tolerating HHP and thermal stresses in *S. cerevisiae*.

### 3.4. Generation of Piezosensitive Mutants from Niigata-Sake Yeast Strain S9arg

#### 3.4.1. Screening and Obtain the Piezosensitive Mutants Derived from Strain S9arg

Based on the relationship between the piezosensitivity and respiration deficiency described above, a screening method for piezosensitive mutant strains was developed. A respiration-deficient mutant was highly likely to show increased piezosensitivity. Respiration-deficient mutants could easily be detected by using a TTC assay [11].

The Niigata-*sake* yeast *S. cerevisiae* strain S9arg produces high levels of ethyl caproate but does not produce urea, and is thus suitable for brewing a high-quality *sake*. We used this strain for producing the prototype of HHP pasteurized sparkling-cloudy *sake*, AWANAMA [5,6]. In the established brewing process, we used HHP treatment at 400 MPa at an ambient temperature for 10 min to inactivate strain S9arg. We thus generated piezosensitive mutants derived from strain S9arg.

After UV irradiation treatment of strain S9arg, the cell suspension was subjected to a TTC staining test. A total of 495 colonies appeared, of which seven colonies were TTC-negative. After subculturing, three TTC-negative isolates were obtained, which were designated as strains UV1, UV4, and UV7. A cell suspension of strain S9arg without UV irradiation treatment was also subjected to the TTC staining test. As a result, two TTC-negative strains were obtained, which were designated as strains NM1 and NM2.

PCR amplification was carried out to analyze the genes in the mitochondrial DNA. For the five mutant strains and strain S9arg, PCR products were amplified using primer sets targeting the 21S rRNA gene and *ORI7*. However, PCR primer sets targeting *COX1* (five primer sets) and *COB* (one primer set) could not produce amplification in the five mutant strains, although they could produce amplification in strain S9arg. These results indicated that in all of these mutants, the *COX1* gene region was deleted as in strain a924E1. Additionally, in the mutants derived from strain S9arg, the *COB* region was also deleted, although only one primer set was used for this analysis.

From these five strains, strains UV1 and NM1 were used for further study. The polyploidy of strains UV1 and NM1 was also analyzed by PCR using primers targeting *MAT* locus. For both strains and the parent strain S9arg, PCR products of *MATa* (544 bp) and *MATα* (404 bp) were observed (data not shown). This result indicated that the mutants were diploid as was strain S9arg.

#### 3.4.2. Pressure Inactivation Characteristics of Strains UV1 and NM1

The pressure inactivation characteristics of the aerobic respiration-deficient mutant strains UV1 and NM1, as well as the parent strain S9arg, were analyzed. The cell suspensions of the strains, obtained after cultivation on a YPD medium at 30 °C for 48 h, were subjected to HHP treatment at 200 MPa at ambient temperature for 0 s, 80 s, and 160 s. After HHP treatment, the viable cells were enumerated (Figure 7). After HHP treatment for 80 s and 160 s, strain S9arg showed no significant loss of viability, compared with the sample subjected to the HHP treatment for 0 s. In contrast, strains UV1 and NM1 showed significant loss of viability, compared with HHP treatment for 0 s. After 160 s of HHP treatment, both strains showed approximately 2-log loss of viability. These results indicated the increased piezosensitivity of strain a924E1. The deficiency in aerobic respiration, caused by the deletion in the *COX1* gene, was shown to be closely related to the increased piezosensitivity.

#### 3.4.3. Laboratory-Scale Sake Brewing Test

To investigate the brewing behavior of strains UV1 and NM1, we carried out a laboratory-scale *sake* brewing test according to a traditional mashing method performed in three stages, called *sandan-jikomi*. Strain S9arg was also used for the brewing test as a control. The CO_2_ generation by ethanol fermentation from the strains during the brewing period was evaluated from the decrease in the weight of the mash, called *moromi*. For the strains UV1 and NM1, as well as the control strain S9arg, *moromi* weight decrements showed comparable increases during the brewing period and showed a tendency to plateau after day 12 (Figure 8.). After 14 d into the brewing period, the *moromi* weight decrements of strains UV1, NM1, and S9arg reached at 40.7 ± 1.7 g, 39.3 ± 0.7 g, and 39.1 ± 1.1 g, respectively. During 14 d brewing period, significant differences were not observed among the *moromi* weight decrements of strains UV1, NM1, and S9arg, with some exceptions: significant differences (*p* < 0.05) were observed between UV1 and S9arg at 5 d and 7 d and NM1 and S9arg at 2 d. This result indicated that the fermentation speed of the piezosensitive mutant strains UV1 and NM1 were comparable with that of the parent strain S9arg. No apparent effects of the deletion in the *COX1* gene on fermentation ability in the brewing period were observed.

To characterization the brewing properties, we analyzed concentrations of ethanol and flavor components in the *moromi* after the laboratory-scale brewing test for 14 d (Table 3). The concentrations of ethanol in the *moromi* created using strains UV1 and NM1 were comparable with that obtained using strain S9arg. The concentrations of the three major flavor components detected, isoamyl acetate, isoamyl alcohol, and ethyl caproate, in the *moromi* obtained using strains UV1 and NM1 did not show significant difference with those obtained using strain S9arg. As a control, concentrations of these compounds in the *moromi* obtained using *S. cerevisiae* strain K7 (Kyokai No. 7), which is a popular *sake* yeast strain, are also shown. The concentrations of isoamyl alcohol and ethyl caproate in the moromi obtained using strain K7 showed significant difference (*p* < 0.05) with those using S9arg. In the *moromi* obtained using strains UV1, NM1, and S9arg, higher concentration of ethyl caproate and lower concentration of isoamyl alcohol were observed compared to those found in the *moromi* obtained using strain K7. These are typical brewing properties of strain S9arg. The mutant strains UV1 and NM1 were shown to have inherited strain S9arg’s brewing properties, but have acquired the increased piezosensitivity.

## 4. Conclusions

In the present study, we analyzed the genome sequence of a piezosensitive mutant *S. cerevisiae* strain a924E1 and compared it with the parent strain KA31a. A large deletion of 10,130 bp within the *COX1* gene, encoding the subunit I of cytochrome *c* oxidase, in the mitochondrial DNA was detected in strain a924E1. The SNPs detected in strain a924E1 were not found to relate to the increased piezosensitivity in strain a924E1. We thus concluded that the increased piezosensitivity in strain a924E1 was caused by the deletion of the *COX1* gene.

A metabolomic approach was employed to demonstrate that strain a924E1 exhibits restricted flux in the TCA cycle. The deletion of the *COX1* gene was expected to lead to limitations in energy generation and in other TCA cycle-dependent metabolic processes, such as amino acid synthesis, resulting in sensitivity to pressure and temperature stresses. The results obtained in this study contribute to a better understanding of the mechanism of HHP inactivation in microorganisms as well as that of tolerance to environmental stresses.

The demonstration that mitochondrial dysfunction caused by the deletion of the *COX1* gene mediates piezosensitivity in *S. cerevisiae* provides a new screening method for the generation of piezosensitive yeast strains. Thus, we obtained piezosensitive mutants from the Niigata-*sake* yeast strain S9arg, based on aerobic respiration-deficiency. The resultant mutant strains, UV1 and NM1, showed increased piezosensitivity compared with strain S9arg. In a laboratory-scale *sake* brewing test, the fermentation behavior of strains UV1 and NM1 were comparable with that of strain S9arg. No apparent effects of the deletion in the *COX1* gene on fermentation ability during brewing were observed. Furthermore, ethanol and flavor component concentrations in the *moromi* obtained using strains UV1 and NM1 were comparable with those obtained using strain S9arg. The mutant strains UV1 and NM1 were shown to have inherited strain S9arg’s brewing properties, but have acquired the increased piezosensitivity.

In this study, molecular and physiological mechanism concerning the piezosensitivity of S. cerevisiae was investigated. Based on the knowledge, we constructed a screening method for generating an increased piezosensitive mutant, while retaining the brewing properties of the parental *sake* yeast strain. This method would provide a new way of applying HHP pasteurization at lower pressure levels.

## Figures and Tables

**Figure 1 foods-10-02247-f001:**
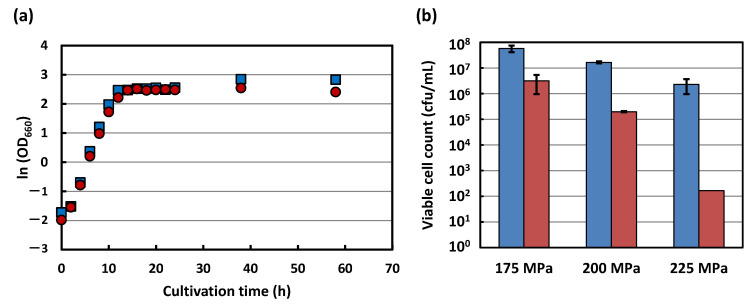
Typical growth curves of strains KA31a (wildtype; blue boxes) and a924E1 (piezosensitive mutant; red circles) (**a**). Cells were grown for 60 h at 30 °C in YPD medium. The time courses of optical densities at 660 nm (OD_660_) are presented as a semi-logarithmic plot. The viable cell count of the strain KA31a (blue boxes) and a924E1 (red boxes) after HHP treatment (**b**). After 48 h cultivations, both strains were subjected to HHP treatment at 175, 200, and 225 MPa at ambient temperature for 360 s.

**Figure 2 foods-10-02247-f002:**
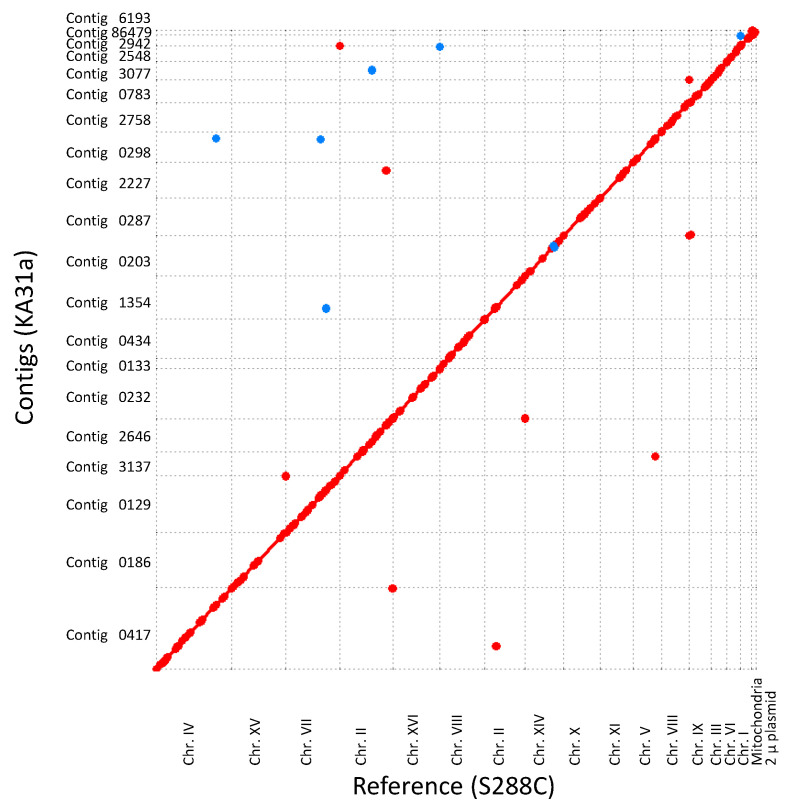
Overall comparison of the genome sequence between strain KA31a and reference strain S288C. Chromosome numbers of the reference strain are shown on the horizontal axis. Contig numbers of strain KA31a are shown on the vertical axis. Conserved regions and complementary regions are shown in red and blue plots, respectively.

**Figure 3 foods-10-02247-f003:**
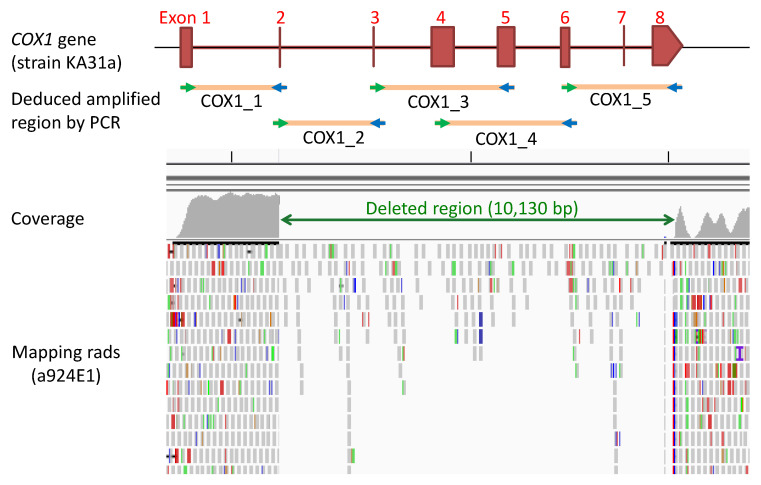
The deleted region in mitochondrial DNA of strain a924E1. Mapping rads of the short reads with coverage are shown. The alignment matches are highlighted in gray and mismatches in different colors (red bars are deletions, blue bars are insertions, green bars are tandem duplications). The map of *COX1* gene in strain KA31a and PCR primer positions with deduced amplified regions are also shown.

**Figure 4 foods-10-02247-f004:**
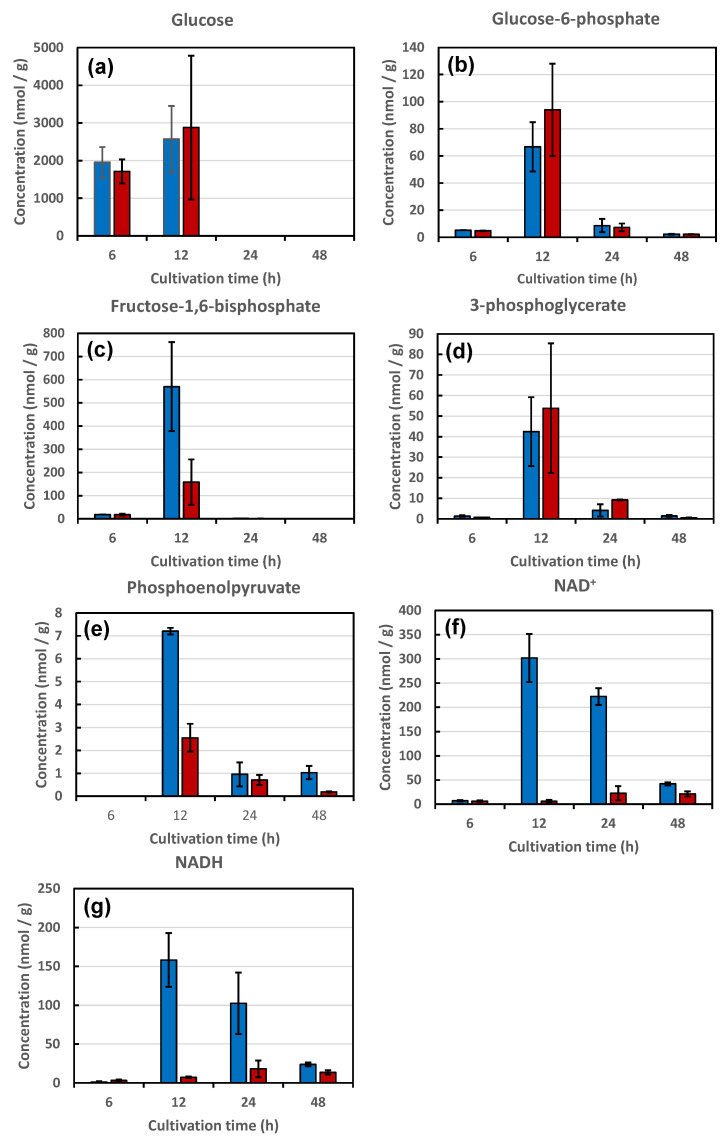
The concentrations of metabolites related to the glycolysis in strains KA31a (open bars) and a924E1 (closed bars) at 6 to 48 h of cultivation. The concentration of each metabolite is shown as nmol of metabolite per gram of dry weight of yeast cells. Panels (**a**–**g**) show the concentrations of glucose, glucose-6-phosphate, fructose-1,6-bisphosphate, 3-phosphoglycerate, phosphoenolpyruvate, NAD^+^, and NADH, respectively. Values are presented as mean ± standard deviation from three independent experiments.

**Figure 5 foods-10-02247-f005:**
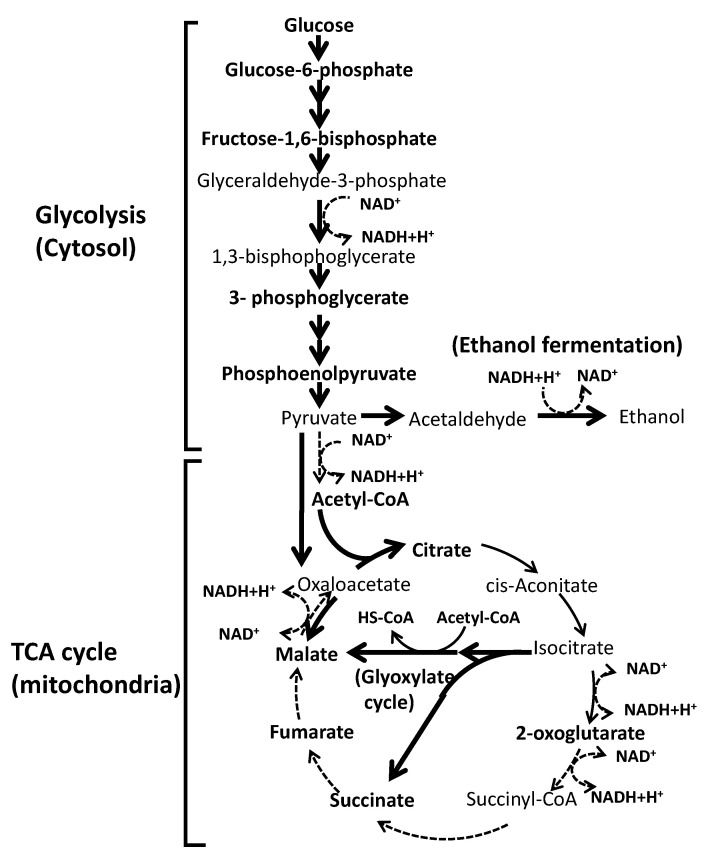
The deduced restricted reaction steps of glycolysis and the TCA cycle based on the present study. The analyzed metabolites are shown in bold letters. The thick solid arrows indicate reaction steps that were comparable between strains KA31a and a924E1. The thin solid arrows indicate reaction steps that were restricted in strain a924E1. The dotted arrows indicate the reaction steps that were almost completely inhibited in strain a924E1.

**Figure 6 foods-10-02247-f006:**
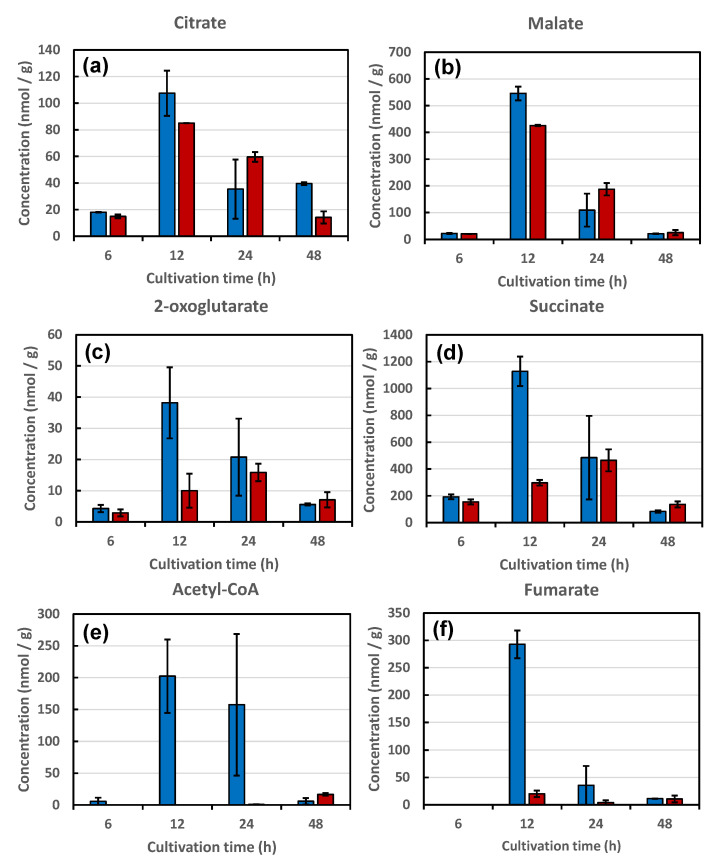
The concentrations of metabolites related to the TCA cycle in strains KA31a (open bars) and a924E1 (closed bars) at 6 to 48 h of cultivation. The concentration of each metabolite is shown as nmol of metabolite per gram of dry weight of yeast cells. Panels (**a**–**f**) show the concentrations of citrate, malate, 2-oxoglutarate, succinate, acetyl-CoA, and fumarate, respectively. Values are presented as mean ± standard deviation from three independent experiments.

**Figure 7 foods-10-02247-f007:**
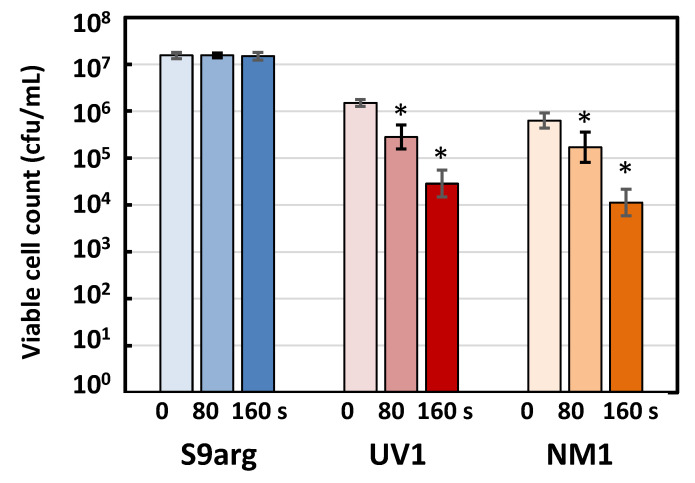
The viable cell counts of strain S9arg and its piezosensitive mutant strains UV1 and NM1 after HHP treatment at 200 MPa at ambient temperature for 0 s, 80 s, and 160 s. Values are presented as mean ± standard deviation from three independent experiments. For each strain, significant difference (*p* < 0.05) of the viable ell count with HHP treatment for 0 s are indicated with asterisks.

**Figure 8 foods-10-02247-f008:**
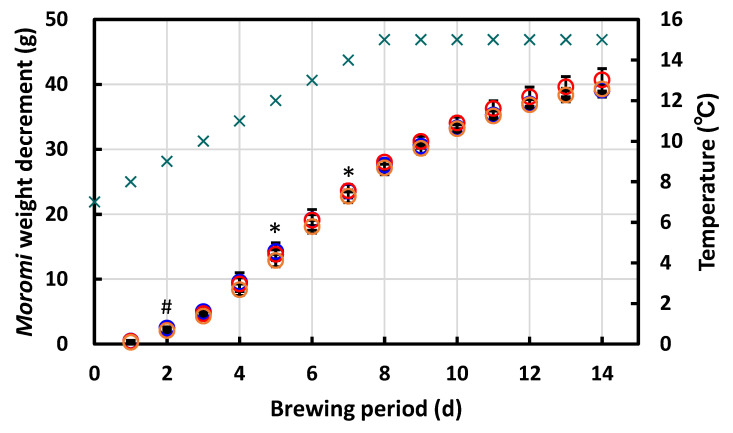
The *moromi* weight decrement during 14 d in a laboratory-scale *sake* brewing test using strains S9arg (blue circles), UV1 (red circles), and NM1 (orange circles). Temperature is also shown in green crosses. Values are presented as mean ± standard deviation from three independent experiments. Significant difference (*p* < 0.05) between UV1 and S9arg (asterisks) and between NM1 and S9arg (number sign) are indicated.

**Table 1 foods-10-02247-t001:** PCR primers targeting the *COX1* gene region.

Primer	Sequence	Product Size (bp)
COX1_1F	5’-TTTTTAGTGGTATGGCAGGAACAG-3’	2576
COX1_1R	5’-TAATACAGCATGACCAACTACTAA-3’	
COX1_2F	5’-TTAGTAGTTGGTCATGCTGTATTA-3’	2546
COX1_2R	5’-ACCTCCAATTAAAGCAGGCATTAC-3’	
COX1_3F	5’-GTAATGCCTGCTTTAATTGGAGGT-3’	3225
COX1_3R	5’-TGCTCTAAGATCTGCATCTAATCC-3’	
COX1_4F	5’-GCTACAGATACAGCATTTCCAAGA-3’	3461
COX1_4R	5’-GTTAGCTAAGGCAACACCAGTTAAA-3’	
COX1_5F	5’-TTTAACTGGTGTTGCCTTAGCTAAC-3’	2978
COX1_5R	5’-AGTGTACAGCTGGTGGAGAAGTTA-3’	

**Table 2 foods-10-02247-t002:** Optimized Q1 and Q3 values for the analyzed metabolites.

Compound	Q1 (m/z)	Q3 (m/z)
Glucose (and Fructose) ^1^	178.9	88.8
Glucose-6-phosphate (and Glucose-1-phosphate) ^2^	258.8	78.9
Fructose-1,6-bisphophate	339.0	78.9
3-phosphoglycerate	184.7	79.0
Phosphoenolpyruvate	166.7	79.0
Acetyl coenzyme A (Acetyl-CoA)	807.6	79.0
Citrate	190.9	86.8
2-oxoglutarate	144.8	101.1
Succinate	116.9	72.9
Fumarate	114.2	72.9
Malate	132.8	114.9
Nicotinamide adenine dinucleotide (NAD^+^)	661.7	539.9
Reduced nicotinamide adenine dinucleotide (NADH)	663.4	78.8
1,4-piperazinediethanesulfonate (PIPES)	300.7	192.9

^1^ Glucose and Fructose could not be distinguished. ^2^ Glucose-6-phosphate and Glucose-1-phosphate also could not be distinguished.

**Table 3 foods-10-02247-t003:** Concentration of ethanol and flavor components in the *moromi* after the laboratory-scale brewing test.

	S9arg	UV1	NM1	K7
Ethanol (%)	12.6 ± 2.3	14.6± 3.5	11.6 ± 2.0	14.5 ± 1.5
Flavor components (mg/L)				
Isoamyl acetate	0.82 ± 0.06	0.70 ± 0.12	0.72 ± 0.12	0.74 ± 0.63
Isoamyl alcohol	68.4 ± 13.6	71.4 ± 7.5	78.0± 12.0	120.7 ± 31.6 *
Ethyl caproate	7.35 ± 3.95	6.94 ± 3.57	4.01 ± 1.22	0.41 ± 0.21 *

Values are presented as mean ± standard deviation from more than three independent experiments. Significant difference (*p* < 0.05) in the values with those using strain S9arg are indicated with asterisks.

## Data Availability

Not applicable.

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
