# Peer review of "Genomic and Metabolomic Analyses of a Piezosensitive Mutant of Saccharomyces cerevisiae and Application for Generation of Piezosensitive Niigata-Sake Yeast Strains"

_foods, 2021, doi:10.3390/foods10102247_

Round 1
Reviewer 1 Report
This manuscript discusses the genomic and metabolomic analyses of a S. cerevisiae mutant sensitive to high pressures. Overall, the paper is well put together. There are some grammar issues and I will point out some (although not all), that are distracting from the science. I would recommend a language editior going through the text.
Minor issues: Line 20,43, 50. The article "the" is missing. There are other places in the text where this was also an issue.
Line 79-84: The phrase "in the study" was used 3 times here. Maybe rewrite the text
Line 203 "to" instead of "for"
Fig 1: for Fig 1a perhaps change one of the "boxes" to another shape. It is helpful for people who do not print out in colour.
Line 285: the word "clearly" is quite strong...just because a PCR does not work does not always mean the gene region is missing. The WGS is very convincing and the PCR substantiate it, but PCRs don't work for a myriad of reasons and this result is also not shown (even in a supplementary) and only mentioned. I will advise to write this section with more caution.
Line 290-291: clumsy sentence...probably "narrowed down to" and not "narrow downed into"
Line 331: sentence is missing a bracket
Line 417: species name not in italics
Line462-469: this section seems more like it should be part of the Materials and Methods
Line 508-509: Figure 8 and its caption separated
Table 3: this table requires some statistical inference. Average +/- standard deviation is not enough
Line 546: clumsy sentence construction
Reviewer 2 Report
Article:
Genomic and metabolomic analyses of a piezosensitive mutant 2 of Saccharomyses cerevisiae and application for generation of 3 piezosensitive Niigata-sake yeast strains.
The article is easy to understand.
I suggest that authors reread the manuscript to correct errors. For example:
line 161 the indication of degrees centigrade is wrong.
other line: species name often not in italics
In all the results shown there is no statistical analysis that confirms the validity of the results obtained. I recommend adding it.
Improves the conclusions by explaining in particular why this avenue of research is valid.
